# Design and Experiment of Air-Suction Maize Seed-Metering Device with Auxiliary Guide

**Li Ding, Yechao Yuan, Yufei Dou, Chenxu Li, Zhan He, Guangmeng Guo, Yi Zhang, Bingjie Chen and He Li \***

College of Mechanical and Electrical Engineering, Henan Agricultural University, Zhengzhou 450002, China;
dingli@henau.edu.cn (L.D.); y95625048@163.com (Y.Y.); m15639927293@163.com (Y.D.);
18238209853@163.com (C.L.); h15737760283@163.com (Z.H.); 13083822967@163.com (G.G.);
zy150104zy@163.com (Y.Z.); cbj334422023@163.com (B.C.)

**\*** Correspondence: chungbuk@163.com

**Abstract:** Due to the irrational design of the seed discharge plate and the vacuum chamber, the high-speed seed filling effect of the air-suction maize precision seed-metering device is poor. Therefore, an air-suction maize seed-metering device with an auxiliary guide is designed to realize high-speed precision seed discharging. An auxiliary guide filling theory is put forward, and the design of the seed plate type hole charging structure is formulated. Fluent 2022 software is used to analyze nine kinds of vacuum chamber structures; the optimal vacuum chamber structure parameters were determined by polar analysis. In order to investigate the changes of negative pressure and flow speed under the dynamic flow field, a slip grid was used to analyze the dynamic flow field with three different operating speeds and negative pressures. It found that the size of negative pressure did not affect the flow field distribution, and the pressure and flow speed gradually decreased as the distance from the inlet was farther away; meanwhile, the negative pressure distribution and air speed distribution were almost unchanged when the holes at different rotational speeds were at the same position. Finally, bench tests were carried out, and three indexes, namely, the qualified index, the multiple index and the missing index, were selected, with operating speed and negative pressure as factors, two-factor five-level orthogonal test was carried out, and the optimal parameter combinations at 6.0, 7.5, 9.0, 10.5, and 12 km/h forward velocity were derived and verified by regression equations. The results showed that the designed seed-metering device was repeated five times when the pressure of the vacuum chamber was −3.5 kPa and the rotational speed of the seed-metering device was 23 r/min, the average grain spacing qualified index was 95.8%, the missing index was 1.6%, the multiple index was 2.6%, and the indexes met the requirements of precision sowing. It is of great significance for our country's seeder to develop in the direction of high-speed and precision.

**Keywords:** maize; air-suction seed-metering device; seed filling; fluent

## 1. Introduction

Precision seed placement technology is an essential tool to achieve conservation of superior seeds, increase crop yields, and reduce production costs [1]. As the core component of precise seed discharge, the performance of the seed dispenser mainly affects the quality of sowing. An air-suction seed dispenser has extensive applicability to seed shape, high precision of single seeding, and meets the requirements of high-speed operation in terms of stability, thus high-speed precision seeding equipment mostly adopts an air-suction seed dispenser mechanism [2,3]. At present, traditional intensive and refined farming has been gradually replaced by machines to meet the requirements of agricultural development in that year, which has led to precision seeding technology appearing in the public vision [4,5].

To address the problem of poor quality high-speed operation of the air-suction seed-metering device, domestic scholars have carried out in-depth research. Tang Han et al. [6] designed a pneumatic type of high-speed maize precision seed-metering device through

a theoretical analysis and bench test, and it is verified that the designed seed catheter can effectively reduce the coefficient of variation. Gao Xiaojun et al. [7,8] designed an air-suction seed dispenser with a new working principle, through theoretical analysis and design, to meet the requirements of high-speed and accurate sowing. Shi Song et al. [9] designed and improved a seed plate with repellent conductivity and verified that the seed plate could play the role of auxiliary seed filling through DEM-CFD and field trials; Wang Guowei et al. [10] designed an air-suction soybean high-speed precision seed-metering device, verified through a simulation analysis and bench test. The designed discharger can complete high-speed operation at low negative pressure; Ding Li et al. [11–13] designed a high-speed precision seed-metering device by using a mechanically assisted seed filling method and verified that the seed plate played a role in assisting seed filling through a simulation and bench test; Liu Rui et al. [14] designed a middle-type seed discharging plate with a perturbing table post, and it was verified through EDEM that its designed seed discharging plate had better performance in assisting seed filling; Xie Dongbo et al. [15] designed a seed plate assisted by perturbing teeth, which improves the seed filling performance of the air-suction seed-metering device; Chen Yulong et al. [16] designed a seed plate with a tilted cam seed pickup structure, which provides theoretical references for the design of a high-speed seed pickup of flat seeds; Su Wei et al. [17] designed a seed plate with flat-band assisted seed filling, which provides theoretical references for the design of a high-speed seed pickup of flat seeds; and in a type of air-suction seed-metering device with a flat belt to assist seed filling, it was verified that the flat belt device could play the role of assisting seed filling through a simulation and bench experiment. The above-mentioned studies can improve the performance of seed filling to a certain extent; however, the main measure is to increase the disturbing structure of the seed and find out how to guide the seed to promote filling and ensure that the negative pressure is constant and stable to carry and the study of this is relatively rare.

The aim of the theoretical analysis of the seed-filling process is to improve the stability of seed filling and seed carrying. With the aid of fluid simulation, the structure of the vacuum chamber is analyzed, and an air-suction maize precision seed discharger is designed. Regression analysis of experimental data has been used to determine an optimum combination of operating parameters.

## 2. Work Process and Theoretical Analysis

### 2.1. Seed-Metering Device Structure and Working Principle

The structure of the auxiliary-guided seed filling air-suction maize precision seed-metering device is shown in Figure 1. It mostly consists of seed plate 1, front shell 2, seed cleaning serration 5, transmission mechanism 6, air sealing cushion 7 and so on.

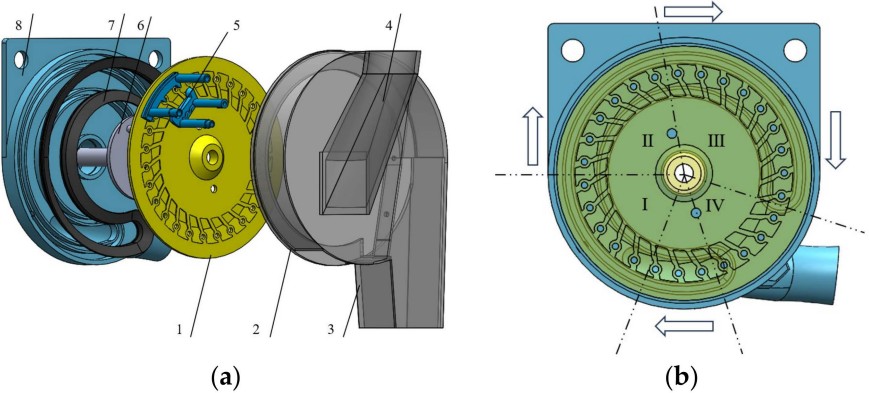

(**a**)  (**b**)

**Figure 1.** Schematic diagram of the structure for the precision seed-metering device. (**a**). Exploded view of the structure. (**b**). Schematic diagram of the work area division. 1. Seed plate; 2. front shell; 3. seed dispenser; 4. seed intake; 5. seed cleaning serration; 6. transmission mechanism; 7. air sealing cushion; and 8. backshell.

According to different regional functions, the seed-metering device is divided into seed-filling area I, seed-clearing area II, seed-carrying area III and seed-unloading area IV, as shown in Figure 1b. In the process of clockwise rotation of the seed plate, the force of the seed is constantly changing. With reference to the empirical division [18], the angle of the I region is set to 50°; the seeds then enters the seed-clearing area. In order to remove the excess seeds and ensure that only one seed is adsorbed in one type of hole, the seed-metering device is set up to clear the seed bilaterally, with a seed-clearing area of 60°; Meanwhile, based on the data analysis of reference [19], the angle of the seed-carrying area is set to 100°, and the angle of the seed-unloading area is set to 60°.

The auxiliary guide seed filling air-suction maize precision seed-metering device working principle is the transmission mechanism with the seed plate rotation, the rear shell and the sealing cushion together constitute a vacuum chamber, and the seed in the negative pressure and the hole guide under the joint action of the charging platform adsorb in the hole. Adsorbed to the hole of the seed with the rotation of the seed plate into the seed-clearing area, the dual side of the clearing and the joint action of the seed sawtooth ensures that the hole of the type of single adsorption corresponds to the nature of the seed. The seed is taken through the seed-carrying area via smooth transport to the seed-unloading area, the negative pressure is cut off, the seed in the gravity centrifugal force under the joint action of the force of the guide hole loading-platform falls into the seed-casting mouth, completing the precision seed-discharge process.

### 2.2. Auxiliary Guided Seed Filling Principle

It is assumed that the material of the seed is uniform, which just enters the seed-filling area. The position of the center of the seed in line with the center on the seeding plate, at this time, is at angle $\alpha = 30°$. The force analysis is shown in Figure 2, and the seed stabilization force equation [20,21] is established as Equation (1).

$$\begin{cases} Q_1 = \sqrt{G^2 + F_C^2 + 2GF_C \cos\alpha} \\ Q = \sqrt{Q_1{}^2 + F_N^2 + 2Q_1 F_N \cos\left(\pi - \theta + \dfrac{\alpha}{2}\right)} \end{cases} \tag{1}$$

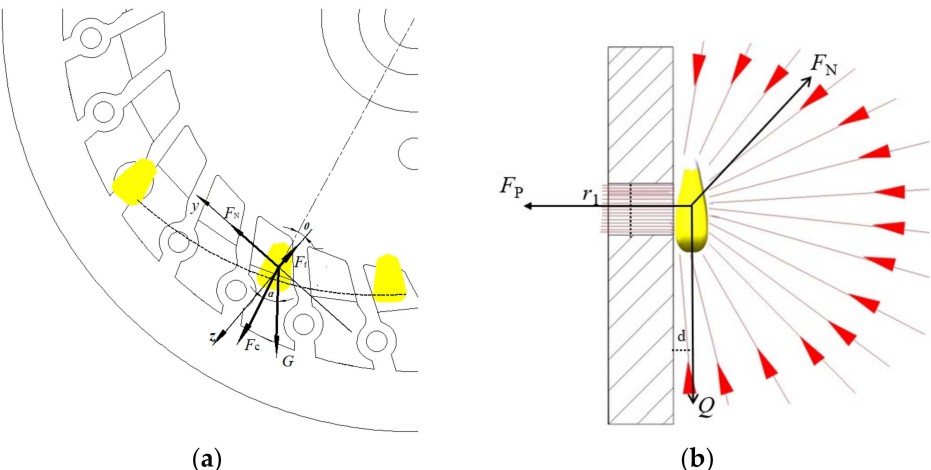

(a)  (b)

**Figure 2.** Schematic of seed-force analysis. (**a**) Motion analysis *y-z* plane for full species. (**b**) Pore-type adsorption *x-z* plane. Note: the yellow part is the motion of the maize, clockwise.

The force analysis of seeded pore adsorption is shown in Figure 2b.

$$\begin{cases} S = \dfrac{\pi}{4}r_1^2 \\ F_\mathrm{P} = PS \\ F_\mathrm{P}\dfrac{r_1}{2} = Qd \end{cases} \tag{2}$$

In the equation, $F_\mathrm{C}$ is the centrifugal force applied to the seed, N; $G$ is the gravitational force on the seed, N; $F_\mathrm{N}$ is the holding force on the seed, N; $\theta$ is the angle in the opposite direction of the centrifugal force $F_\mathrm{C}$ of the seed at the edge of the hole-guiding table, or the angle of the hole-guiding table, (°); $\alpha$ is the angle at which the filling of the seed starts, (°); $Q_1$ is the combined force of gravity and centrifugal force applied to the seed, N; $Q$ is the combined force of gravity, centrifugal force, and holding force applied to the seed, N; $d$ is the distance from $Q$ to the hole-guiding table, mm; $S$ is the area of the suction pore, mm$^2$; $F_\mathrm{P}$ is the adsorption force applied to the seed, N; $r_1$ is the diameter of the suction pore, mm; and $P$ is the adsorption negative pressure of one adsorption pore, Pa.

If the holding force $F_\mathrm{N}$ given to the seed by the pore-conducting table is not taken into account, the negative adsorption pressure $P$ is obtained as follows:

$$P = \frac{8d}{\pi r_1^3}\sqrt{G^2 + F_\mathrm{C}^2 + 2GF_\mathrm{C}\cos\alpha} \tag{3}$$

If the holding force $F_\mathrm{N}$ given to the seed by the pore-conducting charging table is considered, the negative adsorption pressure $P$ is obtained as follows:

$$P = \frac{8d}{\pi r_1^3}\sqrt{Q_1^2 + F_\mathrm{N}^2 + 2Q_1F_\mathrm{N}\cos\left(\pi - \theta + \frac{\alpha}{2}\right)} \tag{4}$$

After the analysis of Equation (4), the $(\alpha/2 - \theta)$ value range is 0~90°, and when other conditions remain unchanged, there is no type of hole-guiding charging table to the seed holding force when the need for adsorption of negative pressure is much greater. Increases in the type of hole-guiding charging table, in the same case for the need for negative pressure decreases, can play a role in auxiliary charging.

In order to reflect the effect of the seed-guided charging, the force analysis of a single seed in the guided charging table [8], is shown in Figure 3.

$$\begin{cases} F_\mathrm{N}\cos\varepsilon = Q\cos\zeta + G\cos\tau \\ F_\mathrm{N}\sin\varepsilon + F_\mathrm{C} + Q\sin\zeta + G\sin\tau \gg F_\mathrm{f} \end{cases} \tag{5}$$

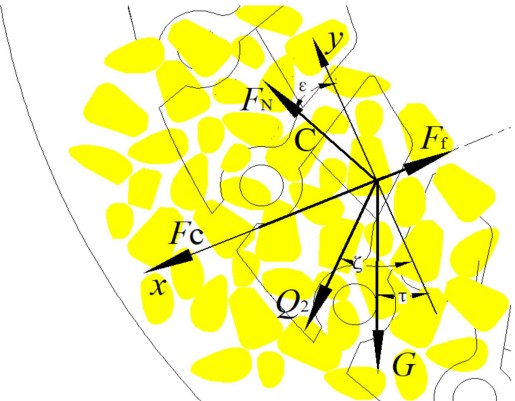

**Figure 3.** Schematic representation of the population state.

In the equation, $\varepsilon$ is the angle between the support force and the *y*-axis, (°); $\zeta$ is the angle between $Q_2$ and the *y*-axis, (°); $\tau$ is the angle between the gravity force and the *y*-axis, (°); and $Q_2$ is the combined force of the population on the stressed seed, N.

When the angle of the guide table is parallel to the line between the hole and the seed plate, $\varepsilon = 0°$, with the increase of the angle $\varepsilon$, $F_N \sin\varepsilon$ in Equation (5) increases, so when $\varepsilon > 0°$, the trend of the seed movement along the *x*-axis is more and more obvious, so the guide table can play a guiding role.

### 2.3. Design of Key Structural Parameters of the Seed Plate

In order to assist the filling and infusion of seeds into the mold hole, the mold hole guide table is designed. Subject to the action of the guide table, so that the seed moves along the tangent direction to the base circle on the guide table, the seed movement and force situation is shown in Figure 4. The speed $V$ of the seed is divided into the speed $V_1$ parallel to the linear speed of the seed plate, and the speed $V_2$ that ensures that the seed can do variable acceleration of linear motion.

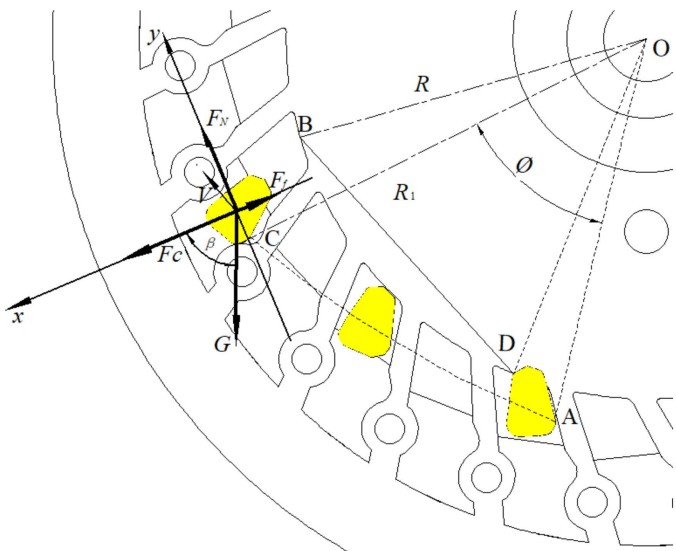

**Figure 4.** Force analysis of seed in seed guide groove.

During the movement of the seed plates from point D to point B, it is necessary to ensure that the seeds move in a variable-speed straight line along the *x*-axis with respect to the seed plates, which is obtained by analyzing the motion of the maize seeds:

$$
\begin{cases}
R_1 = \dfrac{R}{\cos\varphi} \\
V_1 = \omega R_1 \\
V_2 = V_0 + aT \\
V = \sqrt{V_1^2 + V_2^2} \\
\varphi = \omega T \\
a = \dfrac{\omega R \tan\varphi \sec\varphi}{T}
\end{cases}
\tag{6}
$$

Simplifying the above equation yields:

$$
l_{AC} = \int_0^t \sqrt{(\omega R_1)^2 + (V_0 + aT)^2}\, dT = \sqrt{\dfrac{VR\cos\varphi \sin\varphi}{\omega}}
\tag{7}
$$

In the equation, $R_1$ is the distance from point C to the center of the circle, mm; $\omega$ is the rotational speed of the seed plate, rad/s; $V_0$ is the initial speed of the seed, m/s; $V$ is the

absolute speed, m/s; $V_1$ is the implicit speed, m/s; $V_2$ is the relative speed, m/s; $a$ is the acceleration generated by the seed plate on the seed, m/s$^2$; $T$ is the time of the seed plate on the seed action movement, s; $t$ is the seed's actual movement time, s; $\varphi$ is the angle of the line segment OA and OC, rad; $R$ is the radius of the base circle of the seed guide groove curve, mm; and $l_{AC}$ is the absolute displacement distance for the seed movement, mm.

To analyze Equation (6), the absolute displacement distance of the seed in the seed-filling process is approximated as an arc and taken as 48 mm; the range of the actual movement time of the seed is 0~0.52 s when the speed of seed discharging operation is 0~12 km/h, which can be obtained from the force analysis of it:

$$\begin{cases} n_{\mathrm{p}} = \dfrac{60 \times 10^3 v_{\mathrm{m}}}{3.6 S_1 Z} \\ F = F_{\mathrm{C}} + mg \cos \beta - \mu G \sin \beta \\ \omega = 2\pi n_{\mathrm{p}} \\ Ft = mv \end{cases} \tag{8}$$

In the equation, $v_{\mathrm{m}}$ is the operating speed, km/h; $n_{\mathrm{p}}$ is the rotational speed of the seed discharge plate, r/min; $S_1$ is the plant spacing, mm, taken as 250 mm; $Z$ is the number of type holes, taken as 27; $m$ is the mass of a single corn seed, g, taken as 0.35 g; $\beta$ is the angle of $G$ and the $x$-axis, (°); $v$ is the speed of the corn seed in the limiting position, m/s; and $\mu$ is the sliding friction coefficient, according to the reference [22], taken as 0.2.

The simplification yields:

$$\beta = \arcsin \left( \frac{1}{\sqrt{1 + \mu^2}} \right) - \arcsin \left( \frac{\omega^2 r(r - l_{AC})}{g l_{AC}} \right) \tag{9}$$

In the equation, $r$ is the distance from the seed to the center of the circle, mm, because the designed plate radius is 86 mm and the circumferential radius of the type hole is 70 mm; the value range here is 50~72 mm.

According to the general minimum speed $v_{\mathrm{m}}$ of the seeding machine, the minimum speed is 5 km/h and the highest is 12 km/h. Through calculation it can be obtained that the rotational speed $n_{\mathrm{p}}$ is 12.3 r/min and 29.6 r/min, respectively, brought into Equation (9), and it can be obtained that the optimal filling angle $\beta$ of the type hole guide filling table is 40~85°. In order to ensure that the seed filling position is in the optimal range, the simulation will be used for optimization of the structure of the vacuum chamber.

## 3. Structural Analysis of the Flow Field

### 3.1. Simulation Modeling and Analysis

The vacuum chamber fluid region of the air-suction seed-metering device is mainly composed of three parts: vacuum chamber, seed discharging plate and seed filling chamber. According to the flow field analysis results of the air-suction seed-metering device in references [23–25], three factors including vacuum chamber width, vacuum chamber height and air–chamber interface position are selected, and a three-factor three-level orthogonal test is set up and simulation analysis is carried out to derive the optimal parameters, as shown in Figure 5.

A simplified model of the fluid region is drawn in SolidWorks 2020 software, setting a new coordinate system with the center of the seed discharge plate as the center of the circle and saving it in STEP format to export it to SpaceClaim 2022 for image restoration. Meanwhile, the air inlet is defined as the entrance inlet, the lower surface of the seed-filling chamber is defined as the exit outlet, the contact surface of the chamber with the hole is defined as interface1, the contact surface between the hole and the seed filling chamber is defined as interface2, and the rest of the boundary is defined as wall, imported into Ansys-meshing for hexahedral meshing. In the boundary conditions, the inlet pressure of the inlet is set to −3 kPa, the outlet pressure of outlet is set to 0 kPa, the turbulence model

is chosen to be k-epsilon (2 eqn), Realizable model, the wall function is SWF, second-order windward difference format is adopted, and the Coupled algorithm is utilized to solve the problem, defining the maximum number of steps of the operation as 1500, the convergence condition as $10^{-5}$, using hybrid initialization for the operation.

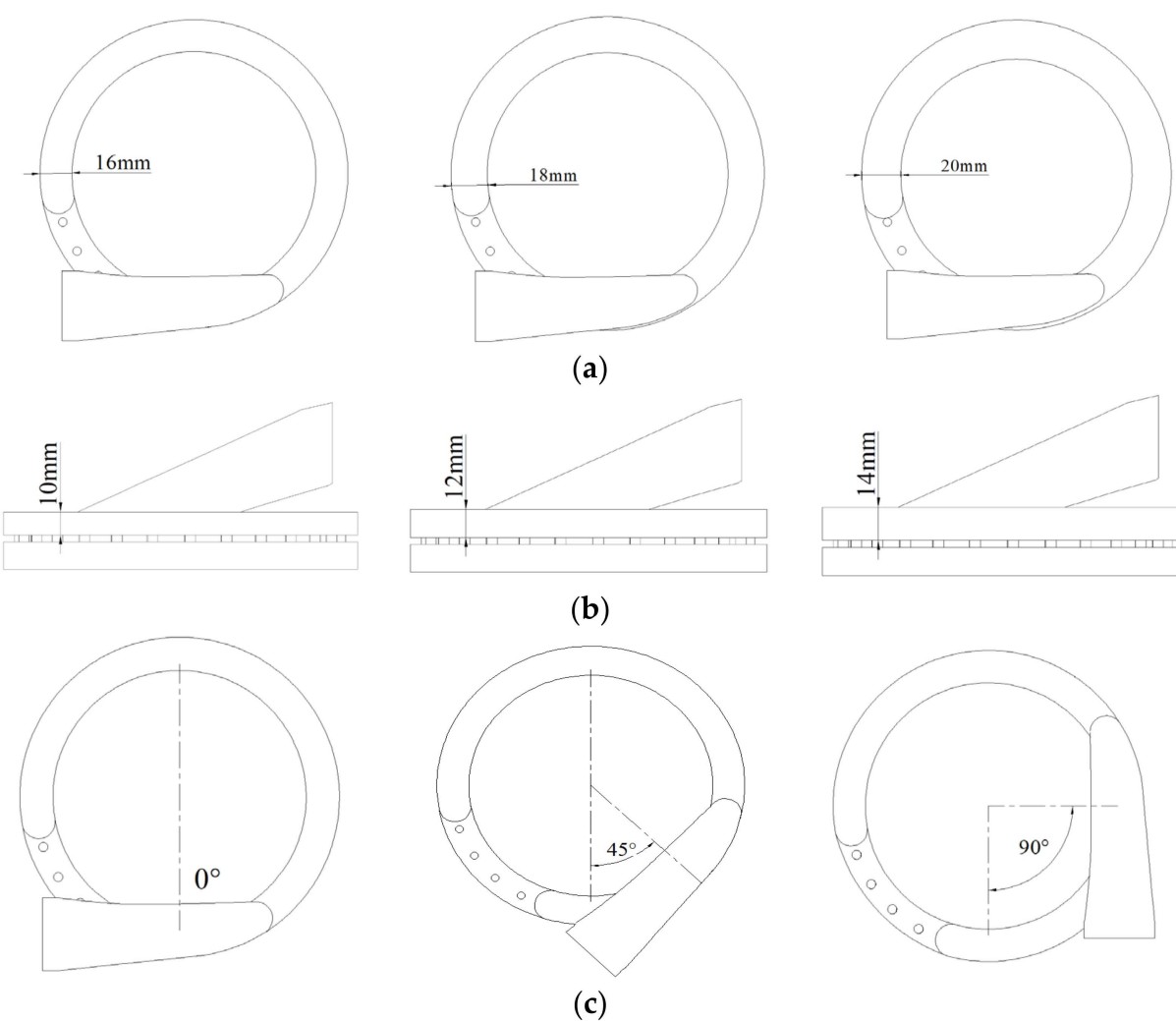

**Figure 5.** Schematic of the levels of factors. (**a**) Vacuum chamber width, (**b**) vacuum chamber height, and (**c**) air–chamber interface position.

*3.2. Simulation and Result Analysis*

In order to reduce the number of simulations, orthogonal test methods are used, and the levels of test factors are shown in Table 1. Fluent 2022 post-processing function is utilized to generate the average value of negative pressure in different regions of the type hole and seed-action surface, and the results are shown in Table 2.

**Table 1.** Experimental factor levels.

| Levels | Experimental Factors | | |
| :---: | :---: | :---: | :---: |
| | Vacuum Chamber Width/(mm) | Vacuum Chamber Height/(mm) | Air–Chamber Interface Position/(°) |
| 1 | 16 | 10 | 0 |
| 2 | 18 | 12 | 45 |
| 3 | 20 | 14 | 90 |

**Table 2.** Results of the three-factor test.

| Test Number | Factors | | | Seed-Filling Area Type Holes Pressure/Pa | Seed-Clearing Area Type Hole Pressure/Pa | Seed-Carrying Area Type Holes Pressure/Pa |
|---|---|---|---|---|---|---|
| | *A* | *B* | *C* | | | |
| 1 | 1 | 1 | 1 | −1730.2 | −768.6 | −307.7 |
| 2 | 1 | 2 | 2 | −1483.7 | −1152.0 | −572.0 |
| 3 | 1 | 3 | 3 | −1213.4 | −1207.7 | −786.2 |
| 4 | 2 | 1 | 2 | −1574.3 | −1222.4 | −558.5 |
| 5 | 2 | 2 | 3 | −1163.3 | −1224.7 | −790.9 |
| 6 | 2 | 3 | 1 | −1732.1 | −1099.4 | −672.8 |
| 7 | 3 | 1 | 3 | −1217.6 | −1242.2 | −781.2 |
| 8 | 3 | 2 | 1 | −1696.8 | −1055.0 | −617.8 |
| 9 | 3 | 3 | 2 | −1455.1 | −1266.8 | −836.0 |

Table 3 is further obtained by polar analysis of variance:

**Table 3.** Extreme variance analysis of simulation test results.

| Evaluation Indexes | Factors | Factor Levels | | | Extremely Poor | Excellent Level |
|---|---|---|---|---|---|---|
| | | **1** | **2** | **3** | | |
| Seed-filling area type holes pressure/Pa | *A* | −1475.8 | −1489.9 | −1456.5 | 33.4 | 2 |
| | *B* | −1507.4 | −1447.9 | −1466.9 | 59.4 | 1 |
| | *C* | −1719.7 | −1504.4 | −1198.1 | 521.6 | 1 |
| Seed-clearing area type holes pressure/Pa | *A* | −1042.8 | −1182.2 | −1188.0 | 145.2 | 3 |
| | *B* | −1077.7 | −1143.9 | −1191.3 | 113.6 | 3 |
| | *C* | −974.3 | −1213.7 | −1224.9 | 250.5 | 3 |
| Seed-carrying area type holes pressure/Pa | *A* | −555.3 | −674.1 | −745.0 | 189.7 | 3 |
| | *B* | −549.1 | −660.2 | −765.0 | 215.9 | 3 |
| | *C* | −532.8 | −655.5 | −786.1 | 253.3 | 3 |

The three factors are not affected to the same degree by polar analysis of variance. For the three metrics of pore pressure in the seed-filled zone, pore pressure in the seed-cleaned zone, and pore pressure in the seed-carrying zone, the largest difference is factor *C*. For the seed-filling area type hole pressure and seed-carrying area type hole pressure, the influence of factor *B* is greater than that of factor *A*, and the degree of influence is from large to small for *C*, *B*, and *A*. For the clearing area type hole pressure, the influence of factor *A* is greater than that of factor *B*, and the degree of influence is from large to small for *C*, *A*, and *B*. In order to avoid missing the sowing phenomenon, the differential pressure at the hole should be as large as possible, which results in the optimal combination of the factors with the largest difference in pressure, and the results are shown in Table 4.

**Table 4.** Optimal combination of factor levels.

| Evaluation Indexes | Optimal Parameter Combination |
|---|---|
| Seed-filling area type holes pressure | $A_2B_1C_1$ |
| Seed-clearing area type holes pressure | $A_3B_3C_3$ |
| Seed-carrying area type holes pressure | $A_3B_3C_3$ |

For the three indicators, according to reference [26], for the air-suction seed-metering device, the most critical work link is the seed filling link; comparing $A_2$ and $A_3$, the two levels of negative pressure in the seed-filling area, the gap is not significant, and the optimal level $A_2$ should be chosen. For the seed-clearing area, the gap between $B_3$ and $B_1$ is not obvious; in order to allow resorption of the seed clearing, $B_1$ should be selected, but for the $C_1$ filling area, the gap between the pressure of the holes in that type is bigger, and

so in order to ensure effective filling, $C_1$ should be selected. For the seed-carrying area, the negative pressure is constant and stable, and from the table it is concluded that the three levels of internal negative pressure are more even and constant, and so $C_1$ should be selected. For the seed-carrying area, the negative pressure that is constant and stable is the most critical; from Table 3, the three levels of internal negative pressure are more constant, and the impact is not significant. Combined with the best filling angle $\beta$ of 40~85° obtained from Equation (8), the best flow structure $A_2B_1C_1$ is selected.

## 4. Simulation and Analysis of Negative Pressure Flow Field

### 4.1. Simulation Pre-Processing and Fluent Parameterization

In order to investigate the negative pressure and flow speed changes under the flow field motion conditions, the slip grid method is used to set the type hole as a dynamic region and the vacuum chamber as a static region, and the grid is re-divided as shown in Figure 6.

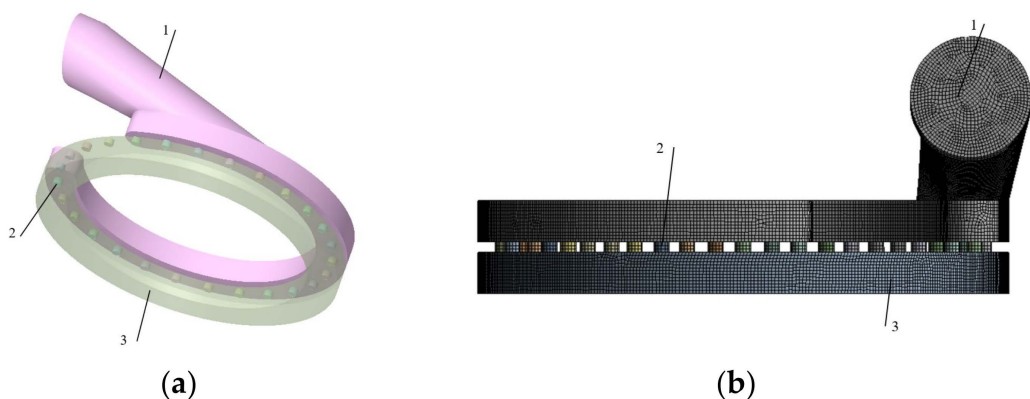

**(a)** **(b)**

**Figure 6.** Three-deminsional model of the flow field and meshing. (**a**) Three-dimensional modeling of the flow field, (**b**) meshing. 1. Vacuum chamber (static area); 2. Type hole (dynamic area); and 3. Seed filling chamber (static area).

### 4.2. Variation of Flow Field with Different Negative Pressure

The pressure distribution and the gas flow rate in the bore have a great influence on the performance of the seed-metering device [27]; the speed of the bore is selected to be analyzed when the negative pressure is −3 kPa, −4 kPa, −5 kPa, and the operating speed is 6 km/h, 9 km/h, and 12 km/h. In order to reduce the number of simulations, the operating speed of 12 km/h is selected for the three negative pressure cases, and the cloud diagrams of three different vacuum levels are obtained as shown in Figure 7.

As can be seen from Figure 7, the absolute value of the pressure under different negative pressure conditions becomes smaller with the distance of the inlet position, and reaches the maximum when the angle with the vertical direction is about 45°, which is exactly located in the range of the β-value of the optimal charging angle of the hole-guided charging table calculated in the previous section, and also verifies the reasonableness of the optimal flow field structure selected in the previous Section 2.2. From Figure 7a, three graphs can be obtained from which we can deduce that the size of the negative pressure cannot make the contact surface of the hole pressure maps change significantly, changes in the size of the negative pressure can only affect the size of the pressure in the vacuum chamber, and cannot affect the distribution of the flow field therein, and the hole connected to the position of the negative pressure value is much higher than the surrounding position. From Figure 7b, it can be seen that the airflow speed is distributed circularly along the axis of the holes, and the speed reaches the maximum value in the position connected with the chamber. In addition, there is gas flow in the seed filling chamber only near the holes, and the airflow speed in the position far away from the holes is almost 0. From Figure 7c,d, it can be seen that the pressure and flow speed gradually decrease the further

away from the position of the air intake port, which is numbered for easy observation, as shown in Figure 8 which shows the numbering of the 27 type holes was completed in a counterclockwise order.

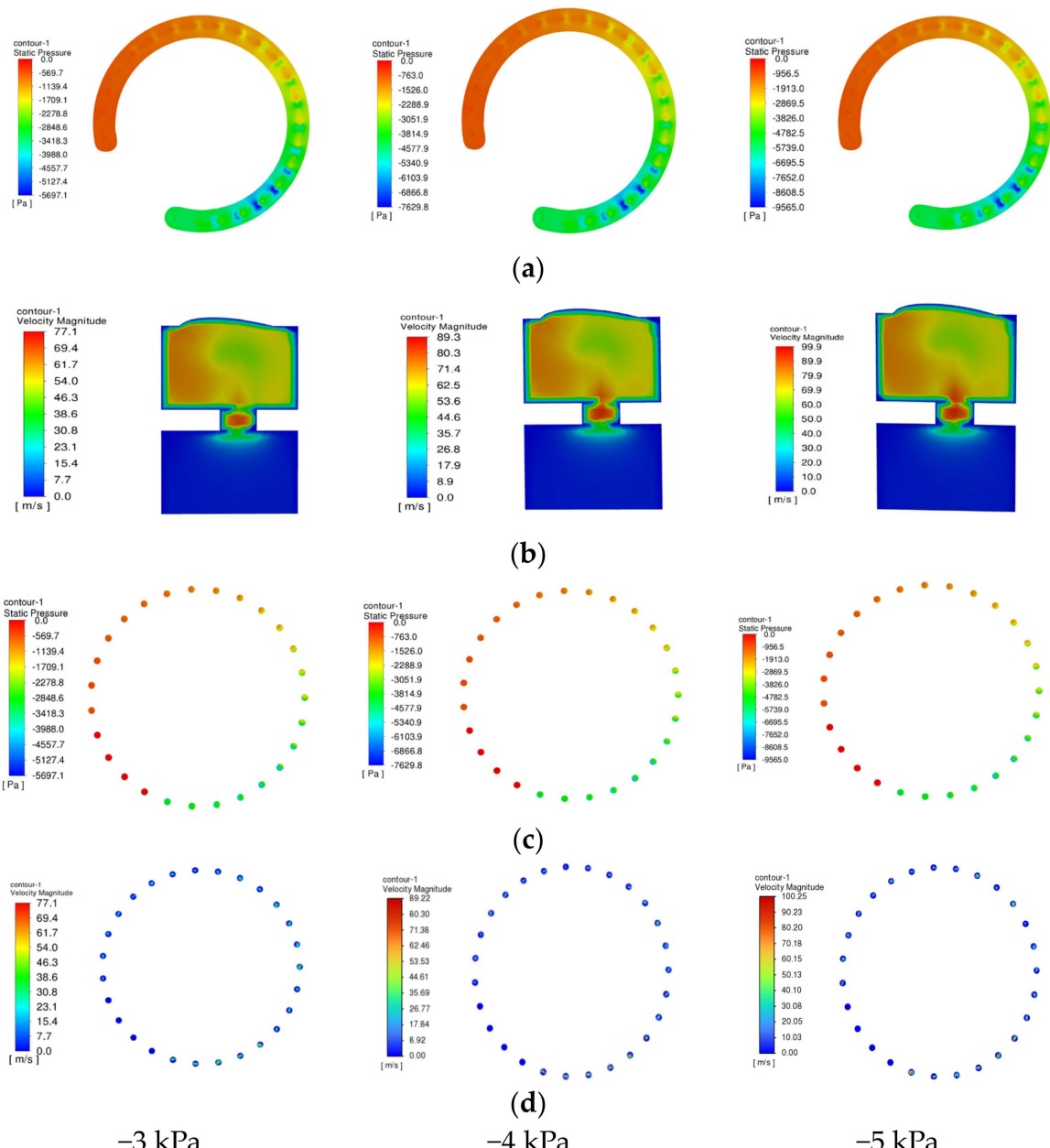

**Figure 7.** Variation of flow field at different vacuum levels. (**a**) Pressure map of the contact surface of the borehole, (**b**) speed maps of end-type holes in the seed-filled region, (**c**) pressure map of the end face of the borehole, and (**d**) speed map of the end face of the borehole.

By using the Report post-processing function in Fluent 2022, the 27 holes follow the speed of the Area-Weighted Average, the average pressure and average flow rate at the 27 hole-shaped cross sections that were obtained and are shown in Figure 9.

As can be seen from Figure 9, the negative pressure and flow speed are the highest in holes No. 5~7, which are located in the range of 40~85° of the optimal charging angle of the hole guide table. The negative pressure and flow speed in holes No. 24~27 are kept at 0 due to their connection to the outside world, while those in holes No. 2~4 are relatively lower due to the reversal of the flow path to the air inlet, and so the speed and flow rate are relatively lower. Overall, the speed and negative pressure values gradually decrease

as movement from the inlet is farther away. The negative pressure and flow rate can be smoothly transitioned to meet the requirement of smooth adsorption.

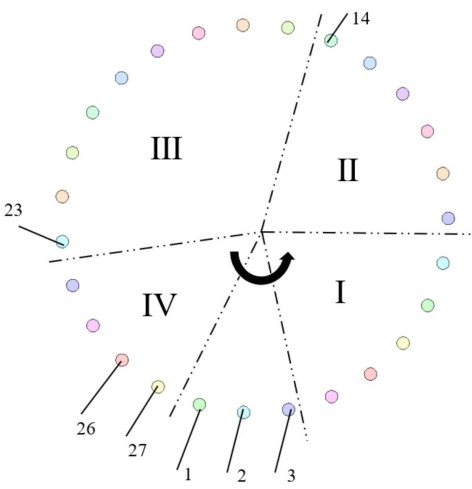

**Figure 8.** Numbering of the holes.

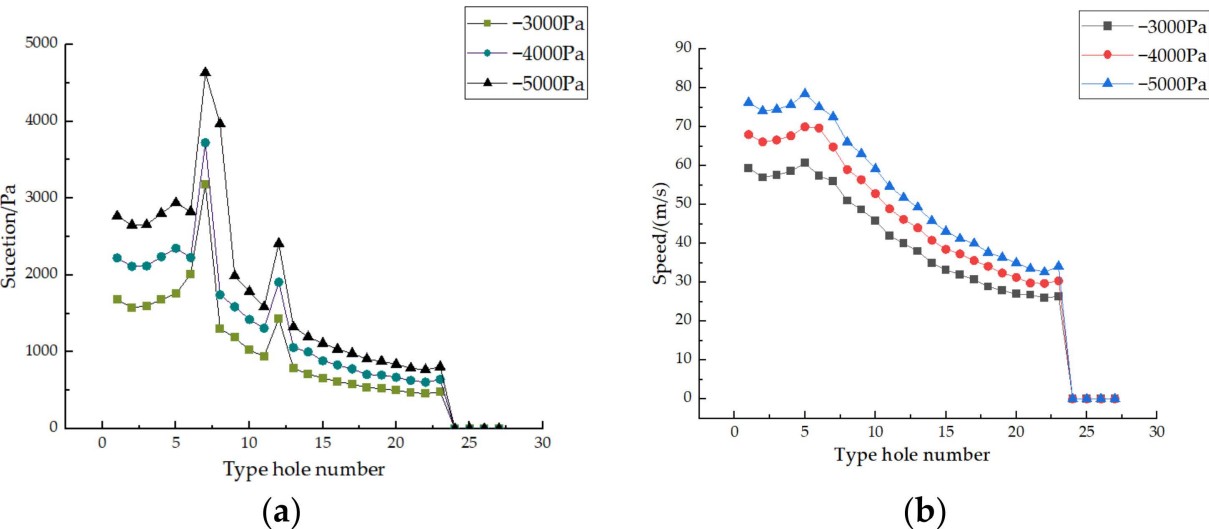

**Figure 9.** Pressure and speed distribution of the molded hole under different negative pressures, (**a**) pressure, (**b**) speed.

### 4.3. Flow Field Changes at Different Speeds

Whether the flow field can meet the adsorption requirements under the change, the rotational speed of the seed plate is crucial. In order to reduce the number of simulations, the simulation is carried out in the case of negative pressure of −3 kPa, and the cloud diagrams under three different forward velocities are obtained as shown in Figure 10.

In order to obtain the specific change value of the speed of the hole in the process of rotation around the center axis, the simulation results of the type holes connected to the vacuum chamber at the cross-section were processed at the time of operational stabilization, where the operational speeds were 6 km/h, 9 km/h and 12 km/h, and the results are shown in Figure 11.

From Figures 10 and 11, it can be seen that there is almost no change in the negative pressure distribution and air speed distribution when the holes are in the same position at different rotational speeds, the rotational speed has almost no effect on the flow field.

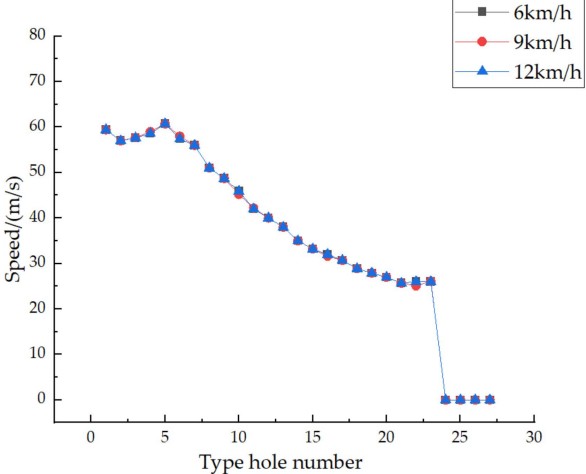

**Figure 10.** Variation of flow field at different rotational speeds. (**a**) Speed profile of borehole end face, (**b**) pressure map of the contact surface of the borehole, and (**c**) speed maps of end-type pores in the seed-filled zone.

**Figure 11.** Distribution of end face speed for different operating conditions.

## 5. Test Program

### 5.1. Test Condition

The test was conducted using a seed dispenser performance tester and stand developed by China Agricultural University, and to make the seed dispenser, it was 3D printed, which is shown in Figure 12 [28]. Maize seed Zhengdan 958 (Shouguang Chunming Seed Shop., Huaifang, China) was selected as the experimental seed.

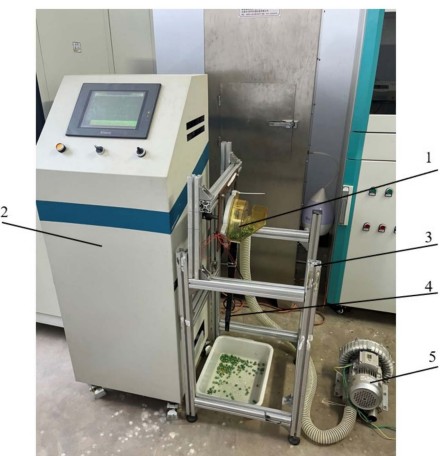

**Figure 12.** Seed displacer performance tester. 1. Air-suction maize seeder (Factory of the Future., Shenzhen, China), 2. seeder performance testing device (China Agricultural University and Langfang Ward Technology Co., Beijing, China), 3. stand, 4. seed guide tube (China Agricultural University and Changzhou Huaiyu Electronics Co., Beijing, China), 5. Shuangfu BDX-400 fan (Dongguan Changan Shuangfeng Hardware Store., Dongguan, China).

According to the single grain (precision) planter test method [29], for each condition repetition of three tests in each test collection of 251 seeds, take the average value of the grain spacing qualified index, multiple index, missing index of the three indexes as a result of the performance evaluation, and the relevant indexes of the calculation equation [30] as follows.

$$\begin{cases} R_1 = \dfrac{n_1}{N_1} \times 100\% \\ R_2 = \dfrac{n_2}{N_1} \times 100\% \\ R_3 = \dfrac{n_3}{N_1} \times 100\% \end{cases} \tag{10}$$

In the equation, $R_1$ is the qualified index, %; $R_2$ is the missing index, %; $R_3$ is the multiple index, %; $n_1$ is the number of qualified seedings; $n_2$ is the number of occurrences of missing; $n_3$ is the number of occurrences of multiple; and $N_1$ is the number of total type holes of the record.

### 5.2. Test Methods

The correspondence between operating speed and rotational speed of the seed plate is shown in Table 5.

**Table 5.** Correspondence between operating speed and rotational speed of the seed plate.

| Speed (km/h) | *B*: Seed Plate Speed (r/min) |
| --- | --- |
| 6.0 | 14.8 |
| 7.5 | 18.5 |
| 9.0 | 22.2 |
| 10.5 | 25.9 |
| 12.0 | 29.6 |

An orthogonal experimental design is carried out with seed plate speed and negative pressure as test factors, and the qualified index, missing index and multiple index as indexes. The coding of experimental factors is shown in Table 6. Finally, the results of the experimental design are shown in Table 7 [31].

**Table 6.** Test factors and Encodings.

| Encodings | B: Seed Plate Speed (r/min) | A: Suction (kPa) |
|---|---|---|
| 1.414 | 14.8 | −3.0 |
| 1 | 18.5 | −3.5 |
| 0 | 22.2 | −4.0 |
| −1 | 25.9 | −4.5 |
| −1.414 | 29.6 | −5.0 |

**Table 7.** Test program and results.

| Encodings | Factors | | Evaluation Indexes | | |
|---|---|---|---|---|---|
| | B: Seed Plate Speed (r/min) | A: Suction (kPa) | Qualified Index/% | Missing Index/% | Multiple Index/% |
| 1 | 0 | 0 | 94.3 | 2.7 | 3.2 |
| 2 | 1 | −1 | 93.2 | 3.2 | 3.6 |
| 3 | 1 | 1 | 90.9 | 8.3 | 0.8 |
| 4 | 0 | 1.414 | 89.7 | 9.4 | 0.9 |
| 5 | −1.414 | 0 | 86.5 | 11.0 | 2.5 |
| 6 | 0 | 0 | 93.2 | 2.2 | 4.6 |
| 7 | −1 | −1 | 91.2 | 4.7 | 4.1 |
| 8 | 0 | −1.414 | 93.8 | 0 | 6.2 |
| 9 | 0 | 0 | 94.2 | 1.6 | 4.2 |
| 10 | 0 | 0 | 93.8 | 2.1 | 4.1 |
| 11 | 1.414 | 0 | 97.9 | 0.9 | 1.2 |
| 12 | 0 | 0 | 94.2 | 1.9 | 3.9 |
| 13 | −1 | 1 | 86.8 | 10.4 | 2.8 |

As can be seen from Table 8, the impact of the two factors of seed plate rotation speed and negative pressure size on the grain spacing qualified index, multiple index and missing index is not the same. The effects of the two factors on the qualified index, missing index and multiple index were all significant, indicating that the designed seed discharger can maintain a high level of performance under different rotational speeds and the aforementioned negative pressure conditions, and that the overall adaptability is strong.

**Table 8.** Analysis of variance of the regression equation.

| Variance Source | $R_1$: Qualified Index | | | | $R_2$: Missing Index | | | | $R_3$: Multiple Index | | | |
|---|---|---|---|---|---|---|---|---|---|---|---|---|
| | Sum of Squares | df | F-Value | p-Value | Sum of Squares | df | F-Value | p-Value | Sum of Squares | df | F-Value | p-Value |
| Model | 104.82 | 5 | 8.29 | ** 0.0074 | 160.69 | 5 | 11.71 | ** 0.0027 | 27.32 | 5 | 14.79 | ** 0.0013 |
| A | 61.73 | 1 | 24.42 | ** 0.0017 | 39.98 | 1 | 14.56 | ** 0.0066 | 2.35 | 1 | 6.37 | * 0.0396 |
| B | 19.53 | 1 | 7.72 | * 0.0273 | 72.56 | 1 | 26.43 | ** 0.0013 | 16.81 | 1 | 45.48 | ** 0.0003 |
| AB | 1.10 | 1 | 0.44 | 0.5301 | 0.090 | 1 | 0.033 | 0.8614 | 0.56 | 1 | 1.52 | 0.2571 |
| $A^2$ | 10.57 | 1 | 4.18 | 0.0802 | 35.41 | 1 | 12.90 | ** 0.0088 | 7.58 | 1 | 20.51 | ** 0.0027 |
| $B^2$ | 14.78 | 1 | 5.85 | * 0.0462 | 18.51 | 1 | 6.74 | * 0.0356 | 0.26 | 1 | 0.71 | 0.4283 |
| Residuals | 17.69 | 7 | | | 19.22 | 7 | | | 2.59 | 7 | | |
| Lack of fit | 16.86 | 3 | 27.02 | ** 0.0041 | 18.56 | 3 | 37.49 | ** 0.0022 | 1.53 | 3 | 1.92 | 0.2678 |
| Error | 0.83 | 4 | | | 0.66 | 4 | | | 1.06 | 4 | | |
| Summation | 122.52 | 12 | | | 179.91 | 12 | | | 29.91 | 12 | | |

Note: * indicates significant (0.01 < *p* < 0.05) and ** indicates highly significant (*p* < 0.01).

### 5.3. Modeling Regression

Based on the test protocol and test results in Table 8, a quadratic polynomial regression model was developed using Design-Expert 10 software for multiple regression fitting analysis of the data with the model equation:

$$
\begin{aligned}
R_1 &= 93.94 + 2.77775A - 1.56228B - 1.4575B^2 \\
R_2 &= 2.1 - 2.23544A + 3.0117B + 2.25625A^2 + 1.63125B^2 \\
R_3 &= 4 - 0.54231A - 1.44942B - 1.04375A^2
\end{aligned}
\tag{11}
$$

Through the matrix plotting of Origin 2021, the response surface plots of each evaluation index with the rotational speed of the seed plate and the magnitude of negative pressure as well as the contour plots are shown in Figure 13.

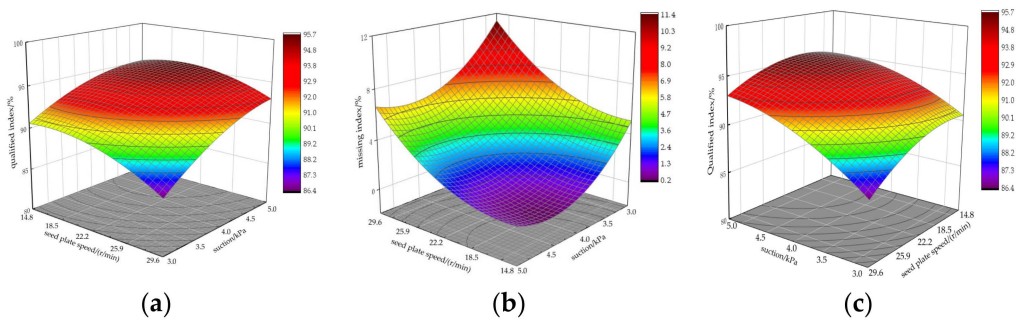

**Figure 13.** Response surface plots for the interaction of factors. (**a**) Impact on qualified index, (**b**) impact on the missing index, and (**c**) impact on multiple index.

By analyzing the response surface plot of qualified index, it can be obtained that when the negative pressure is unchanged, with the increase of seed plate rotational speed, the qualified index of grain spacing presents the trend of first increasing and then decreasing. When the seed plate rotational speed is unchanged, with the increase of negative pressure, the qualified index of grain spacing presents the trend of first increasing and then decreasing. By analyzing the response surface plot of missing index and multiple index, it can be obtained that when the negative pressure is unchanged, with the increase of seed plate rotational speed, the missing index shows the trend of first decreasing and then increasing, and the multiple index presents the trend of first increasing and then decreasing.

### 5.4. Optimal Parameter Optimization

In order to meet the requirement of precision sowing of maize, optimization was carried out by using the optimization numerical in Design-Expert 10 software, and the constraint equations were established as follows with the objectives of maximum of $R_1$, and minimum of $R_2$ and $R_3$, respectively.

$$
\begin{cases}
\max R_1 \\
\min R_2 \\
\min R_3 \\
S.T.\,3.5 \leq A \leq 4.5 \\
18.5 \leq B \leq 29.6
\end{cases}
\tag{12}
$$

The optimal parameter combination was obtained when the size of negative pressure was −3.48 kPa, the rotational speed of the seed-metering device was 23.1 r/min, the qualified index of the seed-metering device was 95.6%, the missing index was 1.5%, and the multiple index was 2.9%. According to the actual working condition, the vacuum chamber pressure was selected as −3.5 kPa, the rotational speed of the seed-metering device was 23 r/min, and the test was repeated five times, and the average grain spacing qualified

index was 95.8%, the missing index was 1.6%, and the multiple index was 2.6%, which met the requirements of precision seeding.

## 6. Discussion

On the basis of previous research, the authors continued to carry out research on the application of the method and equipment for promoting filling of the seed discharges as well as stabilizing the carrying of it. The theory of auxiliary guided seed filling was proposed, and auxiliary guided seed filling discs were designed. The vacuum chamber structure was designed, the negative pressure and flow field changes under different vacuum chamber structures were investigated, and the optimal chamber structure was preferred. The slip grid method was used, which proved that the change of rotational speed had almost no effect on the flow field of the type hole.

In this paper, the bench test was carried out, and the negative pressure range of $-5\sim-3$ kPa and the rotational speed range of 14.8~29.6 r/min were selected for the test. From the response surface map derived from the analysis of the test data, it can be seen that in the range of rotational speed of 14.8~25.9 r/min, the quality of the discharged seeds can reach a higher level, and the auxiliary guiding seed disc can play a role in promoting charging. The optimized flow field structure can make the negative pressure transition smoothly. Optimization of the best seed discharge parameters resulted in the following: the pressure was $-3.5$ kPa, the speed of the seed discharger was 23 r/min, the qualified index was 95.8%, the missing index was 1.6%, and the multiple index was 2.6%. Compared with other seed dischargers on the market, this seed discharger has an auxiliary guided seed filling structure, and the wind pressure demand is lower under high-speed conditions, which can meet the needs of multi-unit coordinated seeding operations. In the later stage of the study, experiments will be conducted in the field to ensure the reliability of the drill operation.

## 7. Conclusions

(1) Aiming at the suction type maize precision seed discharger due to the seed plate and vacuum chamber design is unreasonable, resulting in the seed discharger having a poor high-speed filling effect and poor carry seed stability. An auxiliary guide filling air-suction type maize precision seed-metering devices has been designed, which puts forward the theory of auxiliary guide filling, analyzes the working principle of the auxiliary guide structure, and designs the seed plate hole-guiding structure. The design of the seed plate hole-guiding and filling structure can guide and promote the filling of seeds.

(2) The vacuum chamber structure of the seed-metering device was optimized by Fluent simulation, using a three-factor, three-level orthogonal test, and the vacuum chamber structure was determined to be 18 mm in width, 10 mm in height, and 0° of the gas chamber interface position by polar analysis. In order to investigate the negative pressure and flow rate changes under the flow field movement conditions, the slip grid method was adopted to analyze the changes of the flow field under different negative pressures and rotational speeds, and the negative pressure could only affect the size of the pressure in the chamber. The size of negative pressure can only affect the size of the pressure in the vacuum chamber, and cannot affect the distribution of the flow field. While farther away from the position of the air inlet, the pressure and flow rate gradually reduced, the different speed holes are in the same position when the distribution of negative pressure and the distribution of air speed is almost unchanged, and the rotational speed of the flow field has almost no effect.

(3) Bench tests were carried out using Design-Expert 10 to obtain the response surface plots between the qualified index, missing index, multiple index and speed and pressure, and the optimal combination of operating parameters was obtained. The optimal parameter pressure was selected to be –3.5 kPa, the rotational speed was 23 r/min, and the test was repeated five times, the average qualified index was 95.8%, the missing index was 1.6%, the multiple index was 2.6%, and all the indexes are better than the requirements of the national standard.

## 8. Patents

DING, L.; GUO, C.L.; DOU, Y.F.; XU, Y.F.; CAI, J.J.; LI, H.; SHI, Y.H.; LIU, J.W.; LI, Y.Q. A kind of high-speed air-absorbent seed discharger close to the ground casting fertilizer device: CN202111580362.9 [P]. 17 May 2022.

YANG, L.; DING, L.; LIU, S.R.; YAN, B.X.; HE, X.T. A kind of high-speed precision seed discharger of air suction type with auxiliary seed charging action seed plate: CN201711370047.7 [P]. 13 April 2018.

**Author Contributions:** Conceptualization, L.D. and Y.Y.; methodology, L.D. and Y.Y.; validation, Y.Y., B.C., Z.H. and C.L.; formal analysis, Y.Y., Y.Z., G.G., L.D. and Y.D.; resources, L.D.; data curation, Y.Y.; writing—original draft preparation, Y.Y.; writing—review and editing, L.D. and H.L.; supervision, L.D.; project administration, L.D.; funding acquisition, L.D. All authors have read and agreed to the published version of the manuscript.

**Funding:** This research was supported by National Key Research and Development Program of China (2022YFD2300904), China Agriculture Research System of MOF and MARA (CARS-04), the Key Scientific and Technological Project of Henan Province Department of China (222102110032).

**Institutional Review Board Statement:** Not applicable.

**Data Availability Statement:** The data used to support the results of this study are available from the corresponding authors upon request.

**Acknowledgments:** The authors would like to express their sincere gratitude to their colleges and laboratories and to the reviewers who provided helpful suggestions for this manuscript.

**Conflicts of Interest:** The authors declare no conflicts of interest.

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
