# Peer review of "Design and Experiment of Air-Suction Maize Seed-Metering Device with Auxiliary Guide"

_agriculture, doi:10.3390/agriculture14020169_

Round 1
Reviewer 1 Report
Comments and Suggestions for Authors
The authors have done careful work in the design of the air-suction maize precision seed metering device, with advanced and scientific research methods and reliable results. This study provides support for the design of high-speed corn air-suction precision seeder. and it is recommended to receive and publish the paper.
The following suggestions are for the author's reference.
(1) The design of the seed tray hole guiding structure is mentioned many times in this paper,which can guide and promote the filling of the seeds. However, the structure of the seed tray hole guiding is not given in detail.
(2) CFD-DEM coupling simulation has become a mature method to study gas-solid two-phase flow motion, which can more intuitively reflect the performance of air-suction maize precision seed metering device. Why is only the flow field analysis method used in the simulation?
The following problems still exist in the manuscript.
(1) Figure 1, The annotated letters in Figure (a) are too small and inconsistent with the size of Figure (b).
(2) Figure 4 and formula 7, Point A and point C are not the same point on the corn seed, please explain the position relationship between point A and point C in detail.
(3) Line 181, Please explain in the basis for the range of values for the three factors in detail.
(4) Figure 5, The dimensions marked in Figure (a) are inconsistent, and the model position in Figure (B) is not in order.
(5) Line 235, It is recommended to explain the relation between the best filling angle β and the three factors.
(6) Figure 8, Figure 9, Figure 11, The orientation of all the views is different from the actual working orientation in Figure 1 (b), and it is recommended that these be aligned with it.
(7) Figure 11(c), The dimensions of the flow field cloud image at 9 km/h speed is inconsistent with the other two.
(8) Line 299, There is no clear expression about speed in the text, it is suggested to add
(9) Table 8, The influence of factor (a) and factor (b) on Missing index and Multiple index in the table P-value seems to be opposite to the result discussion.
Author Response
Thank you for your comment and my reply is in this word.

Reviewer 2 Report
Comments and Suggestions for Authors
In this research, an auxiliary guide has been added to maize seed metering device, and numerical analysis has been done to select its characteristics. Then, the technical performance of the optimal mode has been investigated experimentally. There are some concerns regarding the article are announced as follows.
1- The text of the article needs a fundamental revision in terms of writing. Avoid long sentences. Some sentences are confusing, including the first sentence of the abstract.
2- Please check the published international literatures, which are even written by Chinese scholars. Including the following paper.
Tang, H. et al., 2023. Design and test of a pneumatic type of high-speed maize precision seed metering device. Computers and Electronics in Agriculture Volume 211, August 2023, 107997.
3- In the end of introduction, aims section, The sentence is too long. Please, rewrite that clearly.
4- If the additional guide under investigation is the initiative and innovation of this research, it should be highlighted.
5- In Table 8, only for parameter R1, the lack of fit is significant.
6- In equations 11: terms that are not significant should be removed from the regression equation.
7- For the section of “6. Discussion”, the interpretation of the data results of the current work will be presented and compared with the subject In this section, the interpretation of the data results of the current work and comparison with the subject literature should be For this section, the interpretation of the data results of the current work and comparison with the subject literature should be presented.
8- A number of comments have been placed on the file. Please pay attention to them.

Extensive editing of English language required.
Author Response

(The authors gave the same response as above.)

Reviewer 3 Report
Comments and Suggestions for Authors
Li et al. conducted a study and designed an air-suction maize seed-metering device with auxiliary guide to address the issue of poor stability during high-speed seed planting. The paper provides a detailed description of the simulation process and experimental setup, effectively demonstrating the progress achieved in the experiments. This novel seed planting method is intriguing. The designed device can serve as a reference for the design of precision maize seed planters. In my opinion, I recommend the following modifications:
1. Some figures in the manuscript exhibit inconsistent layout and varying image sizes. It is advised to conduct a thorough examination and make appropriate adjustments to ensure consistent figure layout and image sizes throughout the manuscript.
2. There are issues with the English expression of certain technical terms in the manuscript. For instance, the term " seed dispenser" in lines 309 and 310 requires careful review and appropriate modification.
3. According to the experimental methodology outlined in the manuscript, data should be collected for the seed placement indicators of 251 seeds, as 100 seeds are insufficient. It is suggested to supplement the experimental data and perform further analysis.
4. Some of the experimental data in Table 7 only marginally outperforms the national standard, and the experiments were conducted indoors. Therefore, these results cannot support the claim made in line 346 of the manuscript that " the designed seed metering device can maintain a high performance level under different rotational speeds and the above negative pressure conditions, and the overall adaptability is strong."
5. The manuscript contains imprecise language in several instances. For example, the terms "strong versatility" and "low-consumption" mentioned in lines 401 and 402 are not adequately substantiated throughout the text. It is recommended to revise or supplement the relevant experimental evidence to support these claims.
6. Lines 402 to 405 of the manuscript emphasize the contribution of this study to improving the performance of high-speed seed planting operations. If the superiority of this research under high-speed conditions is to be highlighted, it is necessary to conduct tests on the high-speed adaptability of the seed planting device.
Author Response

(The authors gave the same response as above.)

Reviewer 4 Report
Comments and Suggestions for Authors
The work is very interesting, but I have a few tips and comments which I present below:
1. Please indicate chronologically the sowing methods from ancient times to achieving precise sowing? Next, it would be necessary to compare other methods with precision vacuum sowing, indicating the advantages and disadvantages of individual solutions.
• Please describe what share of the total area is the use of the vacuum method? Can you provide statistics?
• Does the use of the vacuum method improve the precision of seed dosing? Can you provide evidence of positive effects on yield quality and quantity?
• The manuscript lacks the geometric dimensions of the device, in particular the sowing disc?
• Please specify in the manuscript how the pressure was controlled (instrument and accuracy)?
• There is extensive literature available in the scope presented, please expand it;
• Please briefly explain the programs used in the article?
• Please describe in detail the seeds used during the research (give dimensions, shape, flatness coefficient, etc.?
The summary does not indicate the future directions of development of this work. Please indicate how the current results have influenced your future research goals.
Author Response

(The authors gave the same response as above.)

Round 2
Reviewer 2 Report
Comments and Suggestions for Authors
The authors have addressed issues and concerns to the extent acceptable in the revised version. It is only necessary to observe the following two points.
1- The first sentence of the abstract is not appropriate. It is enough to observe the points in the same sentence of the first version.
2- The word "we" should be avoided anywhere in the article, including in the objective section.
Minor editing of English language required.
Author Response

(The authors gave the same response as above.)
